# On the Word Boundaries of Emergent Languages Based on Harris's Articulation Scheme

## Abstract

The purpose of this paper is to investigate whether Harris's articulation scheme (HAS) also holds in emergent languages. HAS is thought to be a universal property in natural languages that articulatory boundaries can be obtained from statistical information of phonems alone, without referring to word meanings. Emergent languages are artificial communication protocols that arise between agents in a simulated environment and have been attracting attention in recent years. It is considerd important to study the structure of emergent languages and the similarity to natural languages. In this paper, we employ HAS as an unsupervised word segmentation method and verify whether emergent languages arising from signaling games have meaningful segments. Our experiments showed that the emergent languages arising from signaling games satisfy some preconditions for HAS. However, it was also suggested that the HAS-based segmentation boundaries are not necessarily semantically valid.

## 1 Introduction

Communication protocols emerging among artificial agents in a simulated environment are called *emergent languages* [Lazaridou and Baroni, 2020]. It is important to investigate their structure to recognize and bridge the gap between natural and emergent languages, as several structural gaps have been reported [Kottur et al., 2017, Chaabouni et al., 2019]. For instance, Kottur et al. [2017] pointed out that emergent languages are not necessarily compositional. Such gaps are undesirable because major motivations in this area are to develop interactive AI [Foerster et al., 2016, Mordatch and Abbeel, 2018, Lazaridou et al., 2020] and to simulate the evolution of human language [Kirby, 2001, Graesser et al., 2019, Dagan et al., 2021]. Previous work examined whether emergent languages have the same properties as natural languages, such as compositionality [e.g., Kottur et al., 2017], grammar [van der Wal et al., 2020], entropy minimization [Kharitonov et al., 2020], and Zipf's law of abbreviation (ZLA) [e.g., Chaabouni et al., 2019].[1] *Word segmentation* would be another direction to understand the structure of emergent languages because natural languages not only have construction from word to sentence but also from phoneme to word [Martinet, 1960]. However, previous studies have not gone so far as to address word segmentation, as they treat each symbol in emergent messages as if it were a "word" [Kottur et al., 2017, van der Wal et al., 2020], or ensure that a whole message constructs just one "word" [Chaabouni et al., 2019, Kharitonov et al., 2020].

The purpose of this paper is to study whether *Harris's articulation scheme* (HAS) [Harris, 1955, Tanaka-Ishii, 2021] also holds in emergent languages. HAS is a statistical universal in natural languages. Its basic idea is that we can obtain word segments from the statistical information of phonemes, but without referring to word meanings.[2] HAS can be used for unsupervised word

---

[1]ZLA states that the more frequently a word is used, the shorter it tends to be [Zipf, 1935].

[2]Note that this is different from the famous *distributional* hypothesis [Harris, 1954].

Submitted to 36th Conference on Neural Information Processing Systems (NeurIPS 2022). Do not distribute.

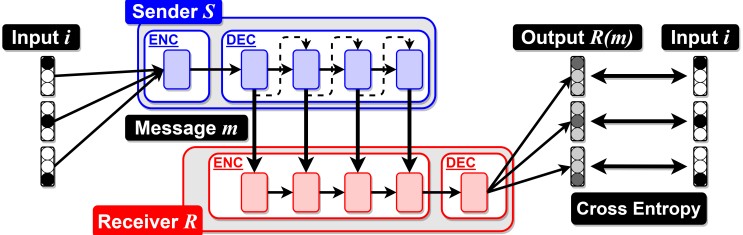

Figure 1: Illustration of a signaling game. Section 3.1 gives its formal definition. In each play, a sender agent obtains an input and converts it to a sequential message. A receiver agent receives the messsage and converts it to an output. Each agent is represented as an encoder-decoder model.

segmentation [Tanaka-Ishii, 2005] to allow us to study the structure of emergent languages. In addition, it should be promising to apply such unsupervised methods, since word segments and meanings are not available beforehand in emergent languages.

The problem is whether emergent languages have meaningful segments. If not, then it means that we find another gap between emergent and natural languages. In this paper, we pose several verifiable questions to answer whether their segments are meaningful.

To simulate the emergence of language, we adopt Lewis's signaling game [Lewis, 1969]. This game involves two agents called *sender* $S$ and *receiver* $R$, and allows only one-way communication from $S$ to $R$. In each play, $S$ obtains an input $i \in \mathcal{I}$ and converts $i$ into a sequential message $m = S(i) \in \mathcal{M}$. Then, $R$ receives $m \in \mathcal{M}$ and predicts the original input. The goal of the game is the correct prediction $R(m) = i$. Figure 1 illustrates the signaling game. Here, we consider the set $\{m \in \mathcal{M} \mid m = S(i)\}_{i \in \mathcal{I}}$ as the dataset of an emergent language, to which the HAS-based *boundary detection* [Tanaka-Ishii, 2005] is applicable. The algorithm yields the *segments* of messages.

Our experimental results showed that emergent languages arising from signaling games satisfy two preconditions for HAS: (i) the conditional entropy (Eq. 2) decreases monotonically and (ii) the branching entropy (Eq. 1) repeatedly falls and rises. However, it was also suggested that the HAS-based boundaries are not necessarily meaningful. Segments divided by the boundaries may not serve as meaning units, while words in natural languages do [Martinet, 1960]. It is left for future work to bridge the gap between emergent and natural languages in terms of HAS, by giving rise to meaningful word boundaries.

## 2 Harris's Articulation Scheme

In the paper "From phoneme to morpheme" [Harris, 1955], Harris hypothesized that word boundaries tend to occur at points where the number of possible successive phonemes reaches a local peak in a given context. Harris [1955] exemplifies the utterance "He's clever" that has the phoneme sequence /hiyzclevər/.[3] The number of possible successors after the first phoneme /h/ is 9: /w,y,i,e,æ,a,ə,o,u/. Next, the number of possible successors after /hi/ increases to 14. Likewise, the number of possible phonemes increases to 29 after /hiy/, stays at 29 after /hiyz/, decreases to 11 after /hiyzk/, decreases to 7 after /hiyzkl/, and so on. Peak numbers are found at /y/, /z/, and /r/, which divides the phoneme sequence into /hiy/+/z/+/klevər/. Thus, the utterance is divided into "He", "s", and "clever".

Harris's hypothesis can be reformulated from an information-theoretic point of view by replacing the *number* of successors with *entropy*. In the following sections, we review the mathematical formulation of the hypothesis as *Harris's articulation scheme* (HAS) and the HAS-based boundary detection [Tanaka-Ishii, 2005]. HAS does involve statistical information of phonemes but does not involve word meanings. This is important because it gives a natural explanation for a well-known linguistic concept called *double articulation* [Martinet, 1960]. Martinet [1960] pointed out that languages have two structures: phonemes (irrelevant to meanings) and meaning units (i.e., words and morphemes). HAS can construct meaning units without referring to meanings.

### 2.1 Mathematical Formulation of Harris's Hypothesis

While Harris [1955] focuses on phonemes for word boundary detection, Tanaka-Ishii [2021] suggests that the hypothesis is also applicable to units other than phonemes. Therefore, in this section, a set

---

[3]There may be other representations for the phonemes, but we follow Harris's notation.

of units is called an *alphabet* $\mathcal{X}$ as a purely mathematical notion that is not restricted to phonemes. Tanaka-Ishii [2005] uses characters for the same purpose. Moreover, Frantzi and Ananiadou [1996] and Tanaka-Ishii and Ishii [2007] investigate the detection of collocation from words.

Let $\mathcal{X}$ be an alphabet and $\mathcal{X}^n$ be the set of all $n$-grams on $\mathcal{X}$. We denote by $X_i$ a random variable of $\mathcal{X}$ indexed by $i$, and by $X_{i:j}$ a random variable sequence from $X_i$ to $X_j$. The formulation by Tanaka-Ishii [2005] involves two kinds of entropy: *branching entropy* and *conditional entropy* [Cover and Thomas, 2006].[4] The *branching entropy of a random variable $X_n$ after a sequence $s = x_0 \cdots x_{n-1} \in \mathcal{X}^n$* is defined as:

$$h(s) \equiv \mathcal{H}(X_n \mid X_{0:n-1} = s) = - \sum_{x \in \mathcal{X}} P(x \mid s) \log_2 P(x \mid s), \qquad (1)$$

where $P(x \mid s) = P(X_n = x \mid X_{0:n-1} = s)$. Intuitively, the branching entropy $h(s)$ means how many elements can occur after $s$ or the uncertainty of the next element after $s$. In addition to $h(s)$, the *conditional entropy of a random variable $X_n$ after an $n$-gram sequence $X_{0:n-1}$* is defined as:

$$H(n) \equiv \mathcal{H}(X_n \mid X_{0:n-1}) = - \sum_{s \in \mathcal{X}^n} P(s) \sum_{x \in \mathcal{X}} P(x \mid s) \log_2 P(x \mid s), \qquad (2)$$

where $P(s) = P(X_{0:n-1} = s)$. The conditional entropy $H(n)$ can be regarded as the mean of $h(s)$ over $n$-gram sequences $s \in \mathcal{X}^n$, since $H(n) = \sum_{s \in \mathcal{X}^n} P(s) h(s)$. $H(n)$ is known to decrease monotonically in natural languages [Bell et al., 1990]. Thus, for a partial sequence $x_{0:n-1} \in \mathcal{X}^n$, $h(x_{0:n-2}) > h(x_{0:n-1})$ holds on average, although $h(s)$ repeatedly falls and rises depending on a specific $s$. Based on such properties, *Harris's articulation scheme* (HAS) is formulated as:[5]

$$\begin{aligned} &\text{If there is some partial sequence } x_{0:n-1} \in \mathcal{X}^n \ (n > 1) \\ &\text{s.t. } h(x_{0:n-2}) < h(x_{0:n-1}), \text{ then } x_n \text{ is at a } \textit{boundary}. \end{aligned} \qquad (3)$$

## 2.2 Boundary Detection Algorithm Based on Harris's Articulation Scheme

In this section, we introduce the HAS-based *boundary detection algorithm* [Tanaka-Ishii, 2005]. Let $s = x_0 \cdots x_{n-1} \in \mathcal{X}^n$. We denote by $s_{i:j}$ its partial sequence $x_i \cdots x_j$. Given $s$ and a parameter *threshold*, the boundary detection algorithm yields boundaries $\mathcal{B}$.[6] It proceeds as follows:

1: $i \leftarrow 0; \ w \leftarrow 1; \ \mathcal{B} \leftarrow \{\}$
2: **while** $i < n$ **do**
3:     Compute $h(s_{i:i+w-1})$
4:     **if** $w > 1$ and $h(s_{i:i+w-1}) - h(s_{i:i+w-2}) > \textit{threshold}$ **then**
5:         $\mathcal{B} \leftarrow \mathcal{B} \cup \{i + w\}$
6:     **end if**
7:     **if** $i + w < n - 1$ **then**
8:         $w \leftarrow w + 1$
9:     **else**
10:         $i \leftarrow i + 1; \ w \leftarrow 1$
11:     **end if**
12: **end while**

Since our targets are emergent languages, the outputs of the boundary detection algorithm do not necessarily mean articulatory boundaries. Instead, we call them *hypothetical boundaries (hypo-boundaries)* and refer to the segments split by hypo-boundaries as *hypo-segments*. Note that there are other similar methods such as Kempe [1999]. We chose Tanaka-Ishii [2005] because it performs well not only for English but also for Chinese, which has many one-character words. Emergent languages might also have such words. With this algorithm, Tanaka-Ishii and Jin [2008] reported F-score $= 83.6\%$ for word boundary detection from phonemes in English and F-score $= 83.8\%$ for word boundary detection from characters in Chinese. They are considerably high scores for unsupervised settings.

---

[4]The term "branching entropy" is from Tanaka-Ishii and Jin [2008], but the definition per se is quite basic.
[5]Although this is called *hypothesis* in Tanaka-Ishii [2005], Tanaka-Ishii and Jin [2006] and Tanaka-Ishii and Ishii [2007], we refer to it as *scheme* following the recent publication [Tanaka-Ishii, 2021].
[6]The original algorithm involves another parameter *maxlen* to ensure $w < \textit{maxlen}$ for practical reasons. We omit it because the message length in emergent languages is fixed in this paper (see Section 3).

## 3 Emergent Language Arising from Signaling Game

We have to define environments, agent architectures, and optimization methods for language emergence simulations. This paper adopts the framework of Chaabouni et al. [2020]. We define an environment in Section 3.1, specify the agent architecture and optimization methods in Section 3.2, and also give an explanation of the compositionality of emergent languages in Section 3.3.

### 3.1 Signaling Game

An environment is formulated based on Lewis's signaling game [Lewis, 1969]. *A signaling game* $G$ consists of a quadruple $(\mathcal{I}, \mathcal{M}, S, R)$, where $\mathcal{I}$ is an *input space*, $\mathcal{M}$ is a *message space*, $S : \mathcal{I} \to \mathcal{M}$ is a *sender agent*, and $R : \mathcal{M} \to \mathcal{I}$ is a *receiver agent*. The goal is the correct reconstruction $i = R(S(i))$ for all $i \in \mathcal{I}$. While the input space $\mathcal{I}$ and the message space $\mathcal{M}$ are fixed, the agents $S, R$ are trained for the goal. An illustration of a signaling game is shown in Figure 1. Following Chaabouni et al. [2020], we define $\mathcal{I}$ as an attribute-value set $\mathcal{D}_{n_{val}}^{n_{att}}$ (defined below) and $\mathcal{M}$ as a set of discrete sequences of fixed length $k$ over a *finite alphabet* $\mathcal{A}$:

$$\mathcal{I} \equiv \mathcal{D}_{n_{val}}^{n_{att}}, \ \mathcal{M} \equiv \mathcal{A}^k = \{a_1 \cdots a_k \mid a_j \in \mathcal{A}\}. \tag{4}$$

**Attribute-Value Set**   Let $n_{att}, n_{val}$ be positive integers called *the number of attributes* and *the number of values*. Then, an *attribute-value set* $\mathcal{D}_{n_{val}}^{n_{att}}$ is the set of ordered tuples defined as follows:

$$\mathcal{D}_{n_{val}}^{n_{att}} = \{(v_1, \ldots, v_{n_{att}}) \mid v_j \in \{1, \ldots, n_{val}\}\}. \tag{5}$$

This is an abstraction of an attribute-value object paradigm [e.g., Kottur et al., 2017] by Chaabouni et al. [2020]. Intuitively, each index $j$ of a vector $(v_1, \ldots v_j, \ldots, v_{n_{att}})$ is an attribute (e.g., *color*), while each $v_j$ is an attribute value (e.g., *blue*, *green*, *red*, and *purple*).[7]

### 3.2 Architecture and Optimization

We follow Chaabouni et al. [2020] as well for the architecture and optimization method.

**Architecture**   Each agent is represented as an encoder-decoder model (Figure 1): the sender decoder and the receiver encoder are based on single-layer GRUs [Cho et al., 2014], while the sender encoder and the receiver decoder are linear functions. Each element $i \in \mathcal{D}_{n_{val}}^{n_{att}}$ has to be vectorized so that it can be fed into or output from the linear functions. Formally, each $i = (v_1, \ldots, v_{n_{att}})$ is converted into the $n_{att} \times n_{val}$-dimensional vector which is the concatenation of $n_{att}$ one-hot representations of $v_j$. During training, the sender samples messages probabilistically. During the test time, it samples them greedily so that it serves as a deterministic function. Similarly, the receiver's output layer, followed by the Softmax, determines $n_{att}$ categorical distributions over values $\{1, \ldots, n_{val}\}$ during training. During the test time, $n_{att}$ values are greedily sampled from the distributions.

**Optimization**   The agents are optimized with the *stochastic computation graph* [Schulman et al., 2015] that is a combination of REINFORCE [Williams, 1992] and standard backpropagation. The sender is optimized with the former, while the receiver is optimized with the latter.

### 3.3 Compositionality of Emergent Languages

An attribute-value set $\mathcal{D}_{n_{val}}^{n_{att}}$ by Chaabouni et al. [2020] is an extension of an attribute-value setting [Kottur et al., 2017] introduced to measure the compositionality of emergent languages. While the concept of compositionality varies from domain to domain, researchers in this area typically regard it as the *disentanglement* of representation learning. Kottur et al. [2017], for instance, set up an environment where objects have two attributes: *color* and *shape*, each of which has several possible values (e.g., *blue*, *red*, ... for color and *circle*, *star*, ... for shape). They assumed that if a language is sufficiently compositional, each message would be a composition of symbols denoting the color value and shape value separately. This concept has been the basis for subsequent studies [Li and Bowling, 2019, Andreas, 2019, Ren et al., 2020, Chaabouni et al., 2020].

---

[7]Although the game is extremely simple, it is suitable to avoid some pitfalls. Lowe et al. [2019] pointed out that agents may not communicate effectively in more complex games than in a signaling game. Bouchacourt and Baroni [2018] suggested that agents fail to capture conceptual properties when $\mathcal{I}$ is a set of images.

**Topographic Similarity** *Topographic Similarity* (TopSim) [Brighton and Kirby, 2006, Lazaridou et al., 2018] is the de facto compositionality measure in emergent communication literature. Suppose we have distance functions $d_{\mathcal{I}}, d_{\mathcal{M}}$ for spaces $\mathcal{I}, \mathcal{M}$, respectively. TopSim is defined as the Spearman correlation between distances $d_{\mathcal{I}}(i_1, i_2)$ and $d_{\mathcal{M}}(S(i_1), S(i_2))$ for all $i_1, i_2 \in \mathcal{I}$ s.t. $i_1 \neq i_2$. This definition reflects an intuition that compositional languages should map similar (resp. dissimilar) inputs to similar (resp. dissimilar) messages. Following previous work using attribute-value objects [e.g., Chaabouni et al., 2020], we define $d_{\mathcal{I}}$ as the Hamming distance and $d_{\mathcal{M}}$ as the edit distance. Because this paper is about message segmentation, we can consider two types of edit distance. One is the "character" edit distance that regards elements $a \in \mathcal{A}$ as symbols. The other is the "word" edit distance that regards hypo-segments as symbols. Let us call the former *C-TopSim* and the latter *W-TopSim*.

# 4    Problem Definition

The purpose of this paper is to study whether Harris's articulation scheme (HAS) also holds in emergent languages. However, this question is too vague to answer. We first divide it into the following:

Q1. Does the conditional entropy $H$ decrease monotonically?

Q2. Does the branching entropy $h$ repeatedly fall and rise?

Q3. Do hypo-boundaries represent meaningful boundaries?

Q3 is the same as the original question, except that Q3 is slightly more formal. However, we have to answer Q1 and Q2 beforehand, because HAS implicitly takes it for granted that $H$ decreases monotonically and $h$ jitters. Although both Q1 and Q2 generally hold in natural languages, neither of them is trivial in emergent languages. Figure 2 illustrates Q1, Q2, and Q3.

Figure 2: Illustration of questions.

It is straightforward to answer Q1 and Q2 as we just need to calculate $H$ and $h$. In contrast, Q3 is still vague to answer, since we do not have prior knowledge about the boundaries of emergent languages and do not even know if they have such boundaries. To mitigate it, we posit the following necessary conditions for Q3. Let $G$ be a game $(\mathcal{D}_{n_{val}}^{n_{att}}, \mathcal{A}^k, S, R)$. If the answer to Q3 is yes, then:

C1. the mean number of hypo-boundaries per message should increase as $n_{att}$ increases,

C2. the size of the vocabulary (set of all hypo-segments) should increase as $n_{val}$ increases,

C3. W-TopSim should be higher than C-TopSim.

**About C1 and C2**    An attribute-value set $\mathcal{D}_{n_{val}}^{n_{att}}$ was originally introduced to measure compositionality. Compositionality, in this context, means how symbols in a message separately denote the components of meaning. In our case, each segment, or *word*, can be thought of as a certain unit that denotes the attribute values, so that the number of words in a message should increase as the corresponding attributes increase. Therefore, if the answer to Q3 is yes, then C1 should be valid. Likewise, the size of the vocabulary should be larger in proportion to the number of values $n_{val}$, motivating C2. Here, we mean by *vocabulary* the set of all hypo-segments. Note that the message length is fixed, because otherwise the number of hypo-segments would be subject to variable message length as well as $(n_{att}, n_{val})$, and the implication of results would be obscure.

**About C3**    C3 comes from the analogy of the linguistic concept called *double articulation* [Martinet, 1960]. In natural languages, meanings are quite arbitrarily related to the phonemes that construct them. In contrast, the meanings are less arbitrarily related to the words. The phonemes do not denote meaning units but the words do. In our case, for example, the attribute-value object (RED, CIRCLE) seems less compositionally related to the character sequence "r,e,d,c,i,r,c,l,e", while it seems more compositionally related to the word sequence "red,circle." This intuition motivates C3.

Based on conditions C1, C2, and C3, Q3 is restated as follows: (Q3-1) Does the mean number of hypo-boundaries per message increase as $n_{att}$ increases?    (Q3-2) Does the vocabulary size increase as $n_{val}$ increases?    (Q3-3) Is W-TopSim higher than C-TopSim?

## 5 Experimental Setup

### 5.1 Parameter Settings

**Input Space** $n_{att}$ and $n_{val}$ have to be varied to answer Q3-1, Q3-2, and Q3-3, while the sizes of the input spaces $|\mathcal{I}| = (n_{val})^{n_{att}}$ must be equal to each other to balance the complexities of games. Therefore, we fix $|\mathcal{I}| = 4096$ and vary $(n_{att}, n_{val})$ as follows:

$$(n_{att}, n_{val}) \in \{(1, 4096), (2, 64), (3, 6), (4, 8), (6, 4), (12, 2)\}. \tag{6}$$

**Message Space** The message length $k$ and alphabet $\mathcal{A}$ have to be determined for a message space $\mathcal{M} = \mathcal{A}^k$. We set $k = 32$, similarly to previous work on *ZLA* [Chaabouni et al., 2019, Rita et al., 2020, Ueda and Washio, 2021] that regards each $a \in \mathcal{A}$ as a "character." Note that $k = 32$ is set much longer than those of previous work on *compositionality* [Chaabouni et al., 2020, Ren et al., 2020, Li and Bowling, 2019] that typically adopts $k \doteq n_{att}$ as if each symbol $a \in \mathcal{A}$ were a "word." We set $\mathcal{A} = \{1, 2, \ldots, 8\}$. Its size $|\mathcal{A}|$ should be as small as possible to avoid the problem of data sparsity when applying boundary detection, and to ensure that each symbol $a \in \mathcal{A}$ serves as a "character." In preliminary experiments, we tested $|\mathcal{A}| \in \{2, 4, 8, 16\}$ and found that learning is stable when $|\mathcal{A}| \geq 8$.

**Architecture and Optimization** We follow Chaabouni et al. [2020] for agent arthitectures and optmization methods. The hidden size of GRU [Cho et al., 2014] is set to 500, following Chaabouni et al. [2020]. All data from an input space $\mathcal{I} = \mathcal{D}_{n_{val}}^{n_{att}}$ are used for training. This dataset is upsampled to 100 times following the default setting of the code of Chaabouni et al. [2020]. The learning rate is set to $0.001$, which also follows Chaabouni et al. [2020]. Based on our preliminary experiments to explore stable learning, a sender $S$ and a receiver $R$ are trained for 200 epochs and the coefficient of the entropy regularizer is set to $0.01$.

**Boundary Detection Algorithm** The boundary detection algorithm involves a parameter *threshold*. Since the appropriate value of *threshold* is unclear, we vary *threshold* as follows:

$$threshold \in \{0, 0.25, 0.5, 0.75, 1, 1.25, 1.5, 1.75, 2\}. \tag{7}$$

### 5.2 Implementation, Number of Trials, and Language Validity

We implemented the code for training agents using the EGG toolkit [Kharitonov et al., 2019].[8] EGG also includes the implementation code of Chaabouni et al. [2020], which we largely refer to. They are published under the MIT license. For now, our code is available on Anonymous GitHub.[9] For each $(n_{att}, n_{val})$ configuration, agents are trained 8 times with different random seeds. Each run took a few hours with a single GPU.[10] In the following sections, an emergent language with a communication success rate of more than $90\%$ is called *a successful language*.

## 6 Results

As a result of training agents, we obtained 7, 8, 6, 8, 7, and 6 successful languages out of 8 runs for configurations $(n_{att}, n_{val}) = (1, 4096), (2, 64), (3, 16), (4, 8), (6, 4)$, and $(12, 2)$, respectively.

### 6.1 Conditional Entropy Monotonically Decreases

To verify Q1, we show the conditional entropy $H(n)$ (Eq. 2) in Figure 3. In Figure 3, the conditional entropies of the successful languages (solid red lines) decrease monotonically. This confirms Q1 in successful languages. Interestingly, the conditional entropies of emergent languages derived from untrained senders do not necessarily decrease, shown as dashed blue lines in Figure 3.[11] The monotonic decrease in conditional entropy emerges after training agents.

---

[8]https://github.com/facebookresearch/EGG

[9]https://anonymous.4open.science/r/HAS-7F4C/

[10]NVIDIA A100.

[11]One might think that the conditional entropy *cannot* increase by its definition. However, this is not the case in our setting (see Appendix A for more details).

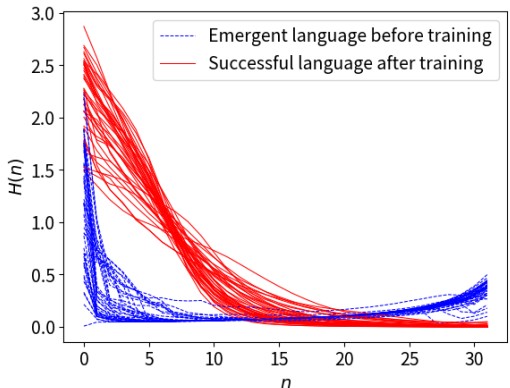
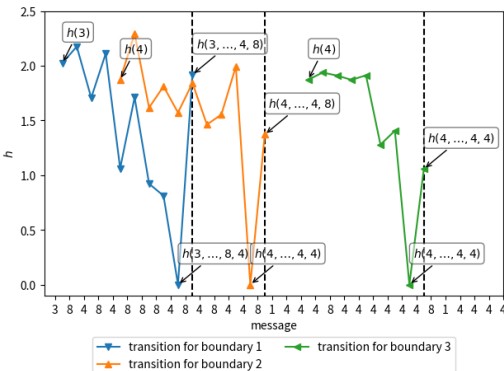

Figure 3: Conditional entropy $H(n)$. Dashed blue lines represent $H(n)$ of languages from untrained agents that finally learned successful languages, while solid red lines represent $H(n)$ of successful languages.

Figure 4: Example transition sequences of the branching entropy $h$ in a message "3,8,4,...,4,4,4" in a successful language for $(n_{att}, n_{val})$ = $(2, 64)$.

## 6.2 Branching Entropy Repeatedly Falls and Rises

Next, to answer Q2, we computed the branching entropy $h(s)$ (Eq. 1) of the successful languages and applied boundary detection. As an example, we show a few actual transitions of $h(s)$ in Figure 4, in which y-axis represents the value of $h(s)$ and x-axis represents a message "3,8,4,...,4,4,4". The message is randomly sampled from a successful language when $(n_{att}, n_{val}) = (2, 64)$. The boundary detection algorithm with *threshold* $= 1$ yields three hypo-boundaries that are represented as dashed black lines in Figure 4. Blue, yellow and green lines with triangle markers represent the transitions of $h(s)$ that yield hypo-boundaries. Note that the $(i + 1)$-th transition of $h(s)$ does not necessarily start from the $i$-th hypo-boundary, due to the definition of the algorithm. For instance, the second transition overlaps the first hypo-boundary. While the conditional entropy decreases monotonically as shown in Figure 3, the branching entropy repeatedly falls and rises in Figure 4. Moreover, we show the mean number of hypo-boundaries per message in Figure 5. Figure 5 indicates that for any $(n_{att}, n_{val})$ configuration, there are hypo-boundaries if *threshold* $< 2$, i.e., the brancing entropy repeatedly falls and rises. These results validate Q2.

## 6.3 Hypo-Boundaries May Not Be Meaningful Boundaries

Next, we investigate whether Q3-1, Q3-2, and Q3-3 hold in successful languages. The results in the following sections falsify all of them. Thus, Q3 may not be true: hypo-boundaries may not be meaningful boundaries.

**Mean Number of Hypo-Boundaries per Message**   See Figure 5 again. The figure shows that the mean number of hypo-boundaries per message does not increase as $n_{att}$ increases. It does not decrease, either. This result falsifies Q3-1. Even when $n_{att} = 1$, there are as many hypo-boundaries as other configurations.

**Vocabulary Size**   Figure 6 shows the mean vocabulary sizes for each $(n_{att}, n_{val})$. The vocabulary size does not increase as $n_{val}$ increases, which falsifies Q3-2. However, focusing on $(n_{att}, n_{val}) \in \{(2, 64), (3, 16), (4, 8), (6, 4)\}$ and $0.25 \le$ *threshold* $\le 1$, there is a weak tendency to support C2. It suggests that hypo-segments are not completely meaningless either.

**C-TopSim vs W-TopSim**   Figure 7 shows C-Topsim and W-Topsim for each $(n_{att}, n_{val})$ and *threshold*.[12] Note that C-TopSim is TopSim with "character" edit distance and W-TopSim is TopSim with "word" edit distance. In Figure 7, *threshold* $= -\infty$ corresponds to C-TopSim, while the others

---

[12]Note that TopSim can only be defined when $n_{att} > 1$.

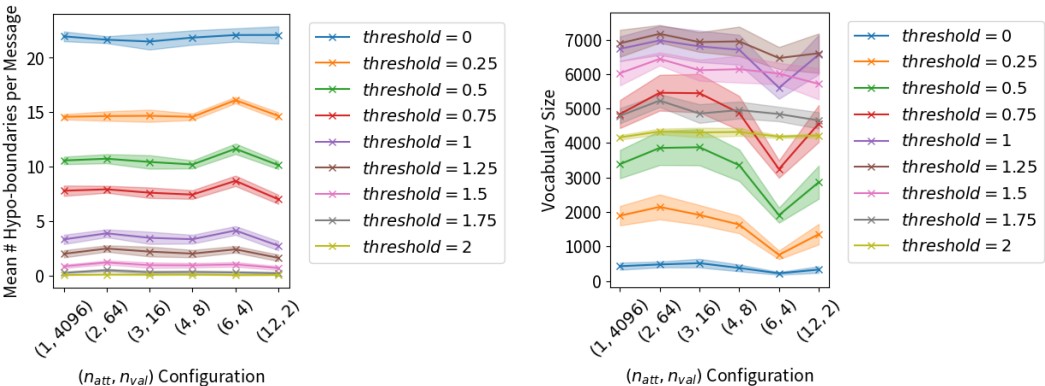

Figure 5: Mean number of hypo-boundaries per message in successful languages. *threshold* varies according to Eq. 7. Each data point is averaged over random seeds and shaded regions represent one standard error of mean (SEM).

Figure 6: Vocabulary size in successful languages. *threshold* varies according to Eq. 7. Each data point is averaged over random seeds and shaded regions represent one SEM.

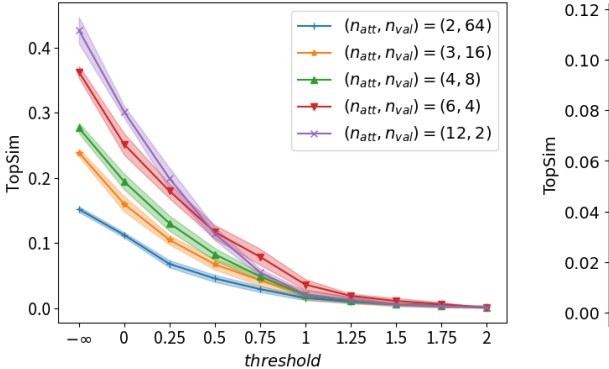

Figure 7: C-TopSim and W-TopSim in successful languages. *threshold* $= -\infty$ corresponds to C-TopSim, while other *threshold* correspond to W-TopSim. Each data point is averaged over random seeds and shaded regions represent one SEM.

Figure 8: hypo-boundary-based W-TopSim compared to random-boundary-based W-TopSim in successful languages for $(n_{att}, n_{val}) = (2, 64)$. Each data point is averaged over random seeds and shaded regions represent one SEM.

correspond to W-TopSim. [13] Our assumption in Q3-3 was C-TopSim $<$ W-TopSim. On the contrary, Figure 7 shows a clear tendency for C-TopSim $>$ W-TopSim, which falsifies Q3-3. Hypo-boundaries may not be meaningful. However, they may not be completely meaningless, either. This is because the hypo-boundary-based W-TopSim is higher than the random-boundary-based W-TopSim in Figure 8. Here, we mean by *random boundaries* the boundaries chosen at random in the same number as hypo-boundaries in each message. Other $(n_{att}, n_{val})$ configurations show similar tendencies (see Appendix B).

### 6.4 Further Investigation: Word Length and Word Frequency

The results so far are related to *compositionality* of emergent languages [e.g., Kottur et al., 2017]. In this section, we further associate our results with previous discussions on *Zipf's law of abbreviation* (ZLA) in emergent languages [Chaabouni et al., 2019, Rita et al., 2020, Ueda and Washio, 2021]. ZLA is known as a statistical property in natural languages that the more frequently a word is used,

---

[13]If we were to apply boundary detection with *threshold* $= -\infty$, it would regard every data point in a message as a boundary. In other words, W-TopSim with *threshold* $= -\infty$ would be identical to C-TopSim. We adopt this notation in order to represent C-TopSim and W-TopSim in a unified manner in a single figure.

the shorter it is [Zipf, 1935]. By considering hypo-segments as "words," we can check whether hypo-segments follow ZLA. Figure 9 shows the hypo-segment lengths sorted by frequency rank for $(n_{att}, n_{val}) = (1, 4096)$.[14] If hypo-segments follow ZLA ideally, they should show a monotonic increase. The distribution of the lengths of the hypo-segments shows a clear ZLA-like tendency for $threshold \in \{0, 0.5\}$, although the tendencies are less clear for the other $threshold$.[15] It means that hypo-segments follow ZLA with an appropriate $threshold$ value. Other $(n_{att}, n_{val})$ configurations show similar tendencies (see Appendix C).

# 7 Discussion

In Section 6.1, we showed that the conditional entropy $H(n)$ decreases monotonically in emergent languages, confirming Q1. In Section 6.2, we demonstrated that the branching entropy $h(s)$ repeatedly falls and rises in emergent languages, which confirms Q2. It is an intriguing result, considering the discussions of Kharitonov et al. [2020], who showed that the entropy decreases to the minimum for successful communication if the message length $k = 1$. In contrast, our results suggest that the (branching) entropy does not simply fall to the minimum when the message length $k$ is longer. However, in Section 6.3, our results indicate that the hypo-boundaries may not be meaningful since Q3-1, Q3-2, and Q3-3 were falsified.

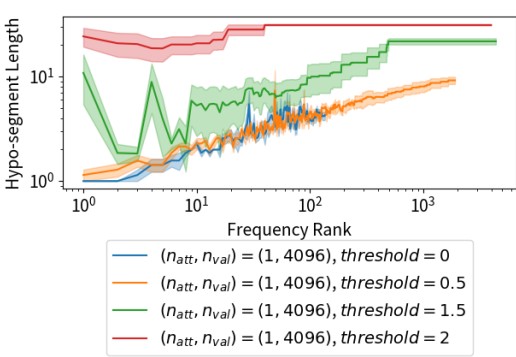

Figure 9: Hypo-segment lengths sorted by frequency rank for $(n_{att}, n_{val}) = (1, 4096)$. Each data point is averaged over random seeds and shaded regions represent one SEM.

Nevertheless, hypo-boundaries may not be completely meaningless either. This is because the hypo-boundary-based W-TopSim is higher than the random-boundary-based W-TopSim. It suggests that HAS-based boundary detection worked to some extent. In addition, the hypo-segments show ZLA-like tendencies with certain $threshold$ values. This is a suggestive result because we neither imposed a length penalty on messages [Chaabouni et al., 2020], modeled the laziness/impatience of agents [Rita et al., 2020], nor modeled short-term memories [Ueda and Washio, 2021]. Of course, it is important to note that it may be just an artifact, analogous to the fact that even a monkey typing sequence divided by the "white space" follows ZLA [Miller, 1957].

This paper showed that there is a gap between emergent and natural languages in terms of word segmentation. There are some potential methods to bridge the gap. For example, several methods have been proposed to facilitate the compositionality of emergent languages, such as iterated learning [Ren et al., 2020], the ease-of-teaching paradigm [Li and Bowling, 2019], and concept game [Mu and Goodman, 2021]. The regularizations for ZLA mentioned above might also help for this purpose. These are left for future work.

# 8 Conclusion

In this paper, we investigated whether Harris's articulation scheme (HAS) also holds in emergent languages. Emergent languages are artificial communication protocols emerging between agents, while HAS is a statistical universal in natural languages. HAS can be used for unsupervised word segmentation. Our experimental results suggest that although emergent languages satisfy some prerequisites for HAS, HAS-based word boundaries may not be meaningful. Our contributions are (1) to focus on the word segmentation of emergent languages, (2) to pose verifiable questions to answer whether emergent languages have meaningful segments, and (3) to show another gap between emergent and natural languages. It is left for future work to bridge the gap between emergent and natural languages in terms of HAS.

---

[14]We picked up only $threshold \in \{0.5, 1.5, 2\}$ and adopted a log-log graph for readability.

[15]The plot shows the zigzagging behavior for $threshold = 1.5$ and most of the hypo-segment lengths hit the message length $k = 32$ for $threshold = 2$.

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
