# Appendix:
# An Attempt to Obtain Word Boundaries of Emergent Languages Based on Harris's Articulation Scheme

## A  Why Can Conditional Entropy Increase in Signaling Game?

One might wonder why the conditional entropy $H(n)$ can increase and think it cannot due to its definition. This is true when we have a single (possibly infinite) sequence. For example, the conditional entropy of an infinite monkey typing sequence is constant since, for any $n \in \mathbb{N}$ and $s \in \mathcal{X}^n$,

$$h(s) = -\sum_{x \in \mathcal{X}} P(x \mid s) \log_2 P(x \mid s) = -\sum_{x \in \mathcal{X}} |\mathcal{X}|^{-1} \log_2 |\mathcal{X}|^{-1} = \log_2 |\mathcal{X}|,$$

$$H(n) = \sum_{s \in \mathcal{X}^n} P(s)h(s) = \log_2 |\mathcal{X}|.$$

Otherwise, $H(n)$ is a weakly decreasing function in a single sequence. However, emergent languages arising from signaling games are not single sequences. Each of them is a set of finite sequences: $L = \{m \in \mathcal{M} \mid m = S(i)\}_{i \in \mathcal{I}}$. Consider, for instance, the following toy language:

$$L_{\text{toy}} = \left\{ \begin{array}{l} aaaaa \\ aaaab \\ aaaac \end{array} \right\}.$$

In $L_{\text{toy}}$, $H(1) < H(4)$ holds, as a symbol after a unigram is most likely to be $a$, while a symbol after a 4-gram is equally likely to be $a$, $b$, and $c$.

 **B   Hypo-boundary-based W-TopSim and random-boundary-based W-TopSim**

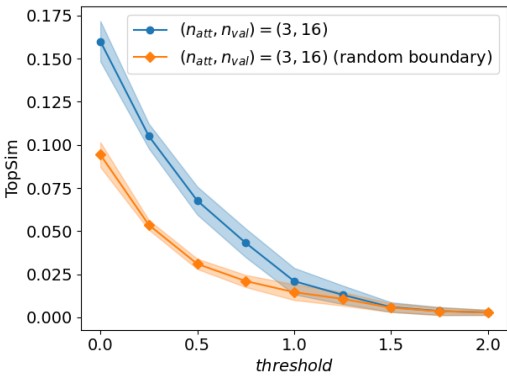

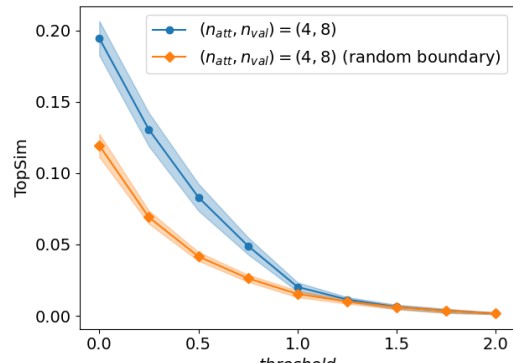

Figure 9: hypo-boundary-based W-TopSim compared to random-boundary-based W-TopSim in successful languages for $(n_{att}, n_{val}) = (3, 16)$. Each data point is averaged over random seeds and shaded regions represent one SEM.

Figure 10: hypo-boundary-based W-TopSim compared to random-boundary-based W-TopSim in successful languages for $(n_{att}, n_{val}) = (4, 8)$. Each data point is averaged over random seeds and shaded regions represent one SEM.

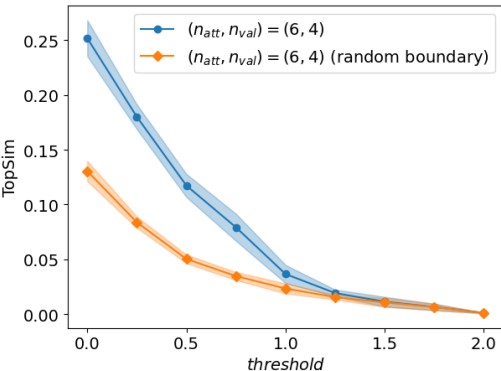

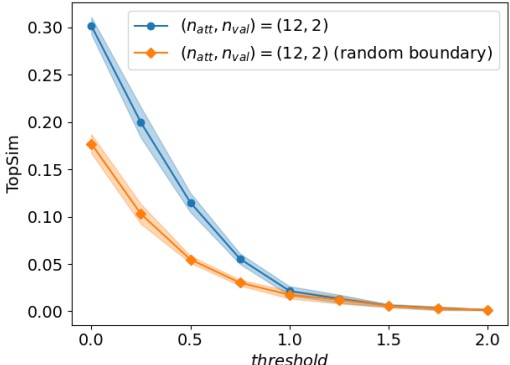

Figure 11: hypo-boundary-based W-TopSim compared to random-boundary-based W-TopSim in successful languages for $(n_{att}, n_{val}) = (6, 4)$. Each data point is averaged over random seeds and shaded regions represent one SEM.

Figure 12: hypo-boundary-based W-TopSim compared to random-boundary-based W-TopSim in successful languages for $(n_{att}, n_{val}) = (12, 2)$. Each data point is averaged over random seeds and shaded regions represent one SEM.

 # C Hypo-segments and Zipf's Law of Abbreviation

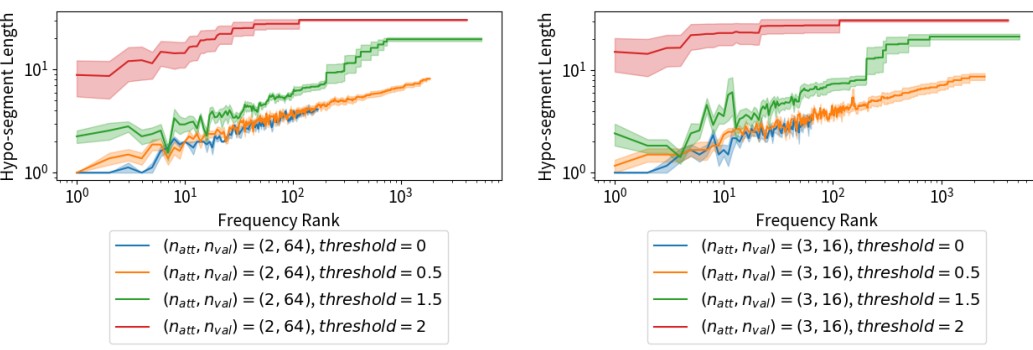

Figure 13: Hypo-segment lengths sorted by frequency rank for $(n_{att}, n_{val}) = (2, 64)$. Each data point is averaged over random seeds and shaded regions represent one SEM.

Figure 14: Hypo-segment lengths sorted by frequency rank for $(n_{att}, n_{val}) = (3, 16)$. Each data point is averaged over random seeds and shaded regions represent one SEM.

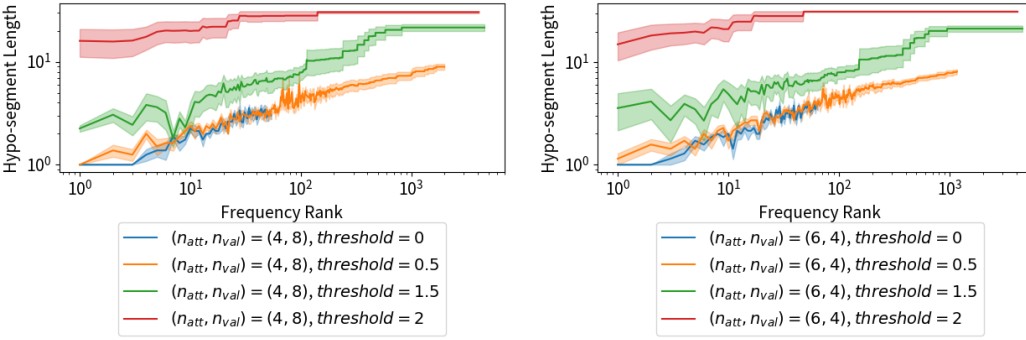

Figure 15: Hypo-segment lengths sorted by frequency rank for $(n_{att}, n_{val}) = (4, 8)$. Each data point is averaged over random seeds and shaded regions represent one SEM.

Figure 16: Hypo-segment lengths sorted by frequency rank for $(n_{att}, n_{val}) = (6, 4)$. Each data point is averaged over random seeds and shaded regions represent one SEM.

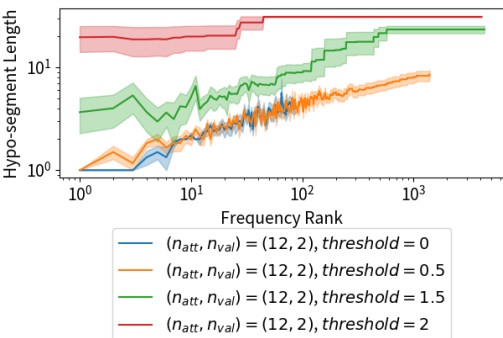

Figure 17: Hypo-segment lengths sorted by frequency rank for $(n_{att}, n_{val}) = (12, 2)$. Each data point is averaged over random seeds and shaded regions represent one SEM.