# OpenReview forum: "On the Word Boundaries of Emergent Languages Based on Harris's Articulation Scheme"
_NeurIPS.cc/2022/Conference — NeurIPS 2022 Submitted_

### Official Review · Reviewer_CeDT · 2022-07-02

**Rating:** 6
**Confidence:** 2
**Soundness:** 3 good
**Presentation:** 4 excellent
**Contribution:** 3 good

**Summary:**

The paper investigates whether HAS also holds in emergent languages, which are languages emerging among artificial agents in a simulated environment. HAS is a universal property in natural languages which says that articulartory boundaries can be obtained from statistical information of phonems without knowing what words mean. The experiments show mixed results---emergent languages satisfy some preconditions for HAS, but the boundaries are not necessarily semantically valid.

**Questions:**

Do you think such results will extrapolate to emergent languages in settings other than Lewis's signaling game. Are there better ways to quantify how "meaningful" the word boundaries may be?

**Limitations:**

Seems fine to me

**Strengths And Weaknesses:**

Strengths:
- Originality: I am not super familiar with this research area but the idea seems quite original. It also seems like a scientifically interesting research topic to me to find the linguistic differences between natural and emergent languages.
- Quality: The experiments seem to be well-motivated, and the background is explained with good attention to details.
- Clarity: The paper writing is quite good.
- Significance: The paper presents extensive experiments. While the experimental results do not fundamentally change the field of NLP, it is a nice focused contribution.

Weaknesses:
- It could be even better to explore emergent languages in multiple settings in addition to Lewis's signaling game.
- Scope of the paper is quite narrow but I do not think this takes away from the merit of the paper.

---

> ### Author Response · Authors · 2022-08-01
> **First Response to Reviewer CeDT**
>
> Thank you for taking the time to read our paper carefully.
>
> Despite your moderate confidence, **your summary accurately captures our claim and your questions are essential for clarifying the significance and limitations of our work.**
>
> # Response to the 1st Question
> > Do you think such results will extrapolate to emergent languages in settings other than Lewis's signaling game.
>
> This is an essential question.
> We must admit that, at present, whether our method applies to more complex settings is not trivial.
> **We clarify this limitation in the camera-ready version.**
>
> The main question in this paper was whether hypothetically obtained segments represent “semantic units” like natural language words.
> That means semantic units have to be explicitly given in our method.
> **It is not evident whether our method is applicable in a more complex setting that does not involve semantic “units” or “composition.”**
> For example, the meaning representation is unclear when agents communicate image data like MSCOCO [Havrylov+, 2017].
> At least, our “C-TopSim vs. W-TopSim” condition might be applicable to such visual settings, if we define Topographic Similarity (TopSim) with cosine (dis)similarity between hidden representations of image data like [Lazaridou+, 2018].
>
> # Response to the 2nd Question
> > Are there better ways to quantify how "meaningful" the word boundaries may be?
>
> This is another crucial question.
> Ground-truth word segmentation data of emergent languages are not given in advance, and it is not even evident whether meaningful word segments exist in the first place.
> **Thus, it is inherently challenging to figure out what makes a suitable evaluation method.**
> As you may intuit, our evaluation method may not be perfect.
> Exploring better ways would be essential for understanding emergent languages.
>
> # Reference
> [Havrylov+, 2017] Serhii Havrylov and Ivan Titov. “Emergence of Language with Multi-agent Games: Learning to Communicate with Sequences of Symbols.” NeurIPS 2017.\
> [Lazaridou+, 2018] Angeliki Lazaridou, Alexander Peysakhovich, and Marco Baroni. “Multi-Agent Cooperation and the Emergence of (Natural) Language” ICLR 2017.

---

### Official Review · Reviewer_RfVg · 2022-07-11

**Rating:** 8
**Confidence:** 4
**Soundness:** 4 excellent
**Presentation:** 4 excellent
**Contribution:** 3 good

**Summary:**

Harris' hypothesis (or articulation scheme, "HAS"; Harris, 1955) suggests that the linguistic boundaries of words can be detected by a count that is unrelated to meaning. However, it is concerned with natural human languages. In this paper, the authors explore if emergent languages between agents share this property with natural languages.

They do this by using HAS for unsupervised word segmentation, i.e., the segmentation of text (in emergent languages) into smaller units. Then, analyzing the results, they ask: Do emergent languages have meaningful segments?

They find that emergent languages somewhat follow HAS, but not in all aspects. Similar to natural languages is that: (1) the conditional entropy decreases monotonically and (2) the branching entropy repeatedly falls and rises. However, in contrast to natural languages, HAS-based boundaries in emergent languages aren't always meaningful. Overall, this is a mixed, but nevertheless interesting result.

**Questions:**

- Do you believe your results are generally applicable to all/most emergent languages? If not, this might be worth clarifying in the relevant passages.
- Except for the above, where do you see the limitations of your work?

**Limitations:**

There are no obvious negative societal impacts of this work. The authors do not explicitly address limitations.

**Strengths And Weaknesses:**

Strengths:
- This is a very clear paper with an easy-to-understand yet interesting research question.
- The experiments are solid and suitable to answer the research question.

Weaknesses:
- The experiments are limited with regards to the settings in which the emergent languages emerge, but the paper is written quite general. This might be worth clarifying.

---

> ### Author Response · Authors · 2022-08-01
> **First Response to Reviewer RfVg**
>
> Thank you very much for your special interest in our work.
> This is encouraging for us.
>
> **Your questions are essential for clarifying the limitation and potential future direction of our study.**
>
> # Response to the 1st Question
> > Do you believe your results are generally applicable to all/most emergent languages? If not, this might be worth clarifying in the relevant passages.
>
> We must admit, at present, that it is not trivial whether our method applies to more complex settings than attribute-value signaling games. **We clarify this limitation in the camera-ready version.**
>
> The main question in this paper is whether hypothetically obtained segments represent semantic units. That means semantic units have to be explicitly given in our method. It is not evident whether our method is applicable in a setting that does not involve semantic units or composition. For example, the meaning representation is unclear when agents communicate image data. At least, our “C-TopSim vs. W-TopSim” condition might be applicable to visual referential games, if we define TopSim with cosine (dis)similarity between hidden representations of image data [Lazaridou+, 2018].
>
> # Response to the 2nd Question
> > Except for the above, where do you see the limitations of your work?
>
> In our signaling game, the input space size is 4096, and the message length is 32.
> That is sufficiently large for a conventional attribute-value setting.
> **However, it is less obvious whether 4096 sentences of length 32 are sufficient for unsupervised word segmentation.**
> We greedily sampled one message per input following the previous work, but it may also be possible to sample multiple messages per input.
>
> # Reference
> [Lazaridou+, 2018] Angeliki Lazaridou, Alexander Peysakhovich, and Marco Baroni. “Multi-Agent Cooperation and the Emergence of (Natural) Language” ICLR 2017.

---

### Official Review · Reviewer_KNRm · 2022-07-11

**Rating:** 5
**Confidence:** 3
**Soundness:** 2 fair
**Presentation:** 3 good
**Contribution:** 2 fair

**Summary:**

This paper presents a new paradigm for evaluating the degree to which artificial emergent languages resemble natural languages. They test the predictions of Harris' articulation scheme (HAS) on the messages produced by converged agents trained in a Lewis signaling game, where a speaker agent is trained by reinforcement learning and a listener agent is trained by supervised learning.
The authors develop a simple word segmentation algorithm designed on the assumptions of HAS. They also propose several predictions which they would expect to hold of the resulting vocabulary / utterance structures based on the nature of the reference game. They find only weak partial support for these predictions in the resulting emergent languages, and claim that this demonstrates a gap between the structure of emergent language and natural language.

**Questions:**

1. What is the baseline / foil for these experiments? The authors claim that HAS is a "statistical universal in natural language," but greedy unsupervised segmentation algorithms based on entropy alone are known to be insufficient for word segmentation tasks / lexicon induction tasks. (The authors cite some sub-100% F-scores themselves in L112-114). This means it's unclear what to expect from this procedure even if we were to stumble upon an emergent language with natural language structure. If humans played the same reference game posed to the communicating agents, what would we expect the outputs of the analysis to look like?
2. What is the intuition behind the ZLA result? This makes sense in systems where the speaker has a length penalty, but not otherwise to me.

**Limitations:**

Yes.

**Strengths And Weaknesses:**

Significance: I am not certain of the significance of the weak negative results of the paper. Some ways that I could imagine this paper would be more significant: 1) if there were a more concretely demonstrated gap with human natural language (see question #1); 2) if there were a positive result (e.g. resulting from implementing one of the ideas in L322-324); 3) if there were adequate qualitative examination (e.g. demonstrating that the TopSim positive result is meaningful, or showing the emergent meanings of hypo-words grounded in the reference game, or showing qualitative examples of compositional utterances).

Clarity: the paper provides *too much* detail, I think -- specifically, I don't see a reason for all of the figures and results to be repeated across different threshold settings. If there is one threshold setting/range that yields intuitive results across analyses, then it'd be best to present just that. If there isn't, that might suggest deeper problems with the procedure! But it seems like 0 < threshold < 0.5 would be sufficient for all the main results?

Quality+originality: Good! Clear and sensible operationalization of abstract ideas in experimental predictions (though see questions).

---

> ### Author Response · Authors · 2022-08-01
> **First Response to Reviewer KNRm (1/2)**
>
> Thank you for giving a thoughtful and detailed review.
> **Your first question is crucial for our work.**
> In response to that question, we tried **a follow-up experiment**.
> We hope that this will mitigate the issue.
>
> # Clarification
>
> First, let us clarify the following just in case there are some misconceptions.
>
> > The authors develop a simple word segmentation algorithm designed on the assumptions of HAS.
>
> This paper does not newly develop but adopts the algorithm of previous work.
> (You may not mean “newly” but just in case.)
>
> > I am not certain of the significance of the weak negative results of the paper.
>
> Our research question is not "How can we obtain high-performance word segmentation in emergent languages?”
> Instead, it is **“Do emergent languages have meaningful segmentation?”**
> We did not obtain a negative result **but we made a scientific discovery**.
>
> > the paper provides too much detail, I think -- specifically, I don't see a reason for all of the figures and results to be repeated across different threshold settings. If there is one threshold setting/range that yields intuitive results across analyses, then it'd be best to present just that. If there isn't, that might suggest deeper problems with the procedure! But it seems like 0 < threshold < 0.5 would be sufficient for all the main results?
>
> We understand what you mean, **but it is difficult to derive a conclusion until we see the full results on threshold values**.
> Specifically, in order to falsify "For some threshold, HAS-based segmentations satisfy the proposed conditions (C1, C2, and C3 in the paper)", we have to show its negation: "For all threshold, HAS-based segmentations do not satisfy the conditions."
> Recall that ground-truth word segmentation data of emergent languages are not given in advance, and it is not even evident whether meaningful word segments exist in the first place.
> That makes it hard to determine the “good” threshold value.
>
> It is, of course, possible to move some of them to appendices, but we believe it is more solid and convincing if all of them are shown together in the main content.
>
> # Response to the 1st Question
>
> [**You can find the follow-up experiment in the next page**]
>
> > If humans played the same reference game posed to the communicating agents, what would we expect the outputs of the analysis to look like?
>
> The abstract nature of the signaling game makes it hard to say precisely how a human would behave.
> Instead, previous work on emergent communication assumes **1) that an ideal language could be decomposed into pieces corresponding to semantic units**, and **2) that this is a minimum property for naturalness**.
> Following this policy, we pose the three conditions, repeated here:
>
> - C1. the mean number of hypo-boundaries per message should increase as $n_{att}$  increases,
> - C2. the size of the vocabulary (set of all hypo-segments) should increase as $n_{val}$ increases,
> - C3. W-TopSim should be higher than C-TopSim.
>
> > What is the baseline / foil for these experiments? The authors claim that HAS is a "statistical universal in natural language," but greedy unsupervised segmentation algorithms based on entropy alone are known to be insufficient for word segmentation tasks / lexicon induction tasks.
>
> **We believe it practical to apply simple and interpretable methods like HAS at first**, in particular because emergent languages do not even necessarily have meaningful boundaries.
> HAS is simple, deterministic, and easy-to-interpret, while linguistically motivated.
>
> **However**, as you point out, our paper does not show whether C1-3 are satisfied in ideal languages (probably what you call baseline/foil).
> That makes you wonder if the HAS-based algorithm is sufficient for word segmentation.
> **Therefore, we have tried the follow-up experiment and confirm that C1-3 are indeed satisfied in ideal languages** (see the next page).
>
> **We will include this result in the camera-ready.**
>
> # Response to the 2nd Question
> > What is the intuition behind the ZLA result? This makes sense in systems where the speaker has a length penalty, but not otherwise to me.
>
> George Zipf hypothesizes that ZLA comes from a trade-off between efficiency and accuracy. On the other hand, it has also been mathematically proven [Miller, 1957] that segments split by “white space” obey Zipf's law and ZLA, even if they are monkey-typing sequences. **The present result is more likely to be a mathematical consequence of the latter rather than from efficiency pressure**.
>
> # Reference
> [Miller, 1957] George A Miller. “Some Effects of Intermittent Silence.” The American Journal of Psychology 70, no. 2 (1957).

---

> > ### Author Response · Authors · 2022-08-01
> > **First Response to Reviewer KNRm (2/2)**
> >
> > # Follow-up Experiment in Response to the First Question
> > ## Motivation
> > - To show that
> >   1. the HAS-based algorithm work for an ideal language,
> >   2. conditions C1-C3 are satisfied in the ideal language,
> >   3. and therefore, the HAS-based algorithm is sufficient for word segmentation in our case.
> > - Here, the ideal language should be decomposed into segments corresponding to semantic units.
> >
> > ## Setup
> > - Recall that $n_{att}$ is the number of attributes, $n_{val}$ is the number of values, an attribute-value set $D_{n_{att}}^{n_{val}}$ is a meaning space, and $(n_{att},n_{val})\in\lbrace(1,4096),(2,64),(3,16),(4,8),(6,4),(12,2)\rbrace$.
> > - Set the alphabet size $|\mathcal{A}|=8$ (same as the paper). Set the message length $k=36$ (slightly different from the paper, but has a good property in the sense that $k\textrm{ mod }n_{att}=0$ for all $n_{att}$).
> > - Make an ideal synthetic language in the following procedure:
> >   - Make a random segment $s_{j,v_j}$ that denotes each attribute-value meaning unit $v_j$.
> >   - Ensure that all the segment lengths are $k/n_{att}$.
> >   - Make a synthetic message that denotes each $(v_1,\ldots,v_{n_{att}})\in D_{n_{att}}^{n_{val}}$, by concatenating the corresponding segments $s_{j,v_j}$.
> >   - Collect the synthetic messages to form a synthetic language.
> > - Then, apply the HAS-based algorithm to the synthetic language. Here we set $\textrm{threshold}=0.5$.
> > - Obtain 8 languages for each $(n_{att},n_{val})$ with different random seeds.
> >
> > ## Result
> > | $n_{att}$ | $n_{val}$ | # boundaries | vocabulary size | C-TopSim | W-TopSim |
> > | :--: | :--: | :--: | :--: | :--: | :--: |
> > | $1$ | $4096$ | $1.52$ ($\pm 0.01$) | $9407.1$ ($\pm 37.8$) | (undefined) | (undefined) |
> > | $2$ | $64$ | $2.68$ ($\pm 0.03$) | $394.8$ ($\pm 7.9$) | $0.134$ ($\pm 0.001$) | $0.232$ ($\pm 0.002$) |
> > | $3$ | $16$ | $3.64$ ($\pm 0.15$) | $96.8$ ($\pm 3.2$) | $0.201$ ($\pm 0.005$) | $0.465$ ($\pm 0.010$) |
> > | $4$ | $8$   | $4.91$ ($\pm 0.19$) | $62.2$ ($\pm 2.6$) | $0.227$ ($\pm 0.009$) | $0.627$ ($\pm 0.021$) |
> > | $6$ | $4$   | $7.02$ ($\pm 0.19$) | $39.7$ ($\pm 1.3$) | $0.253$ ($\pm 0.013$) | $0.763$ ($\pm 0.015$) |
> > | $12$ | $2$ | $11.31$ ($\pm 0.43$) | $28.2$ ($\pm 0.8$) | $0.256$ ($\pm 0.048$) | $0.862$ ($\pm 0.009$) |
> >
> > - ($\pm\cdot$) represents the standard error.
> > - According to the table, **All of C1-3 are satisfied in the synthetic languages.**
> > - **Therefore, the HAS-based algorithm is sufficient for word segmentation in our setting.**

---

> > > ### Comment · Reviewer_KNRm · 2022-08-03
> > > **Response to authors**
> > >
> > > Hi authors, thanks for your detailed response and follow up experiment -- this is great!
> > >
> > > I was concerned (but did not write earlier) that the failure of C1 may be due to the fact that the speaker agent is forced to output a fixed-length message, regardless of the amount of information necessary to convey for a particular game. Thus the HAS algorithm picks up on hypo-boundaries of speaker outputs that may be meaningless to the listener but are there due to the output length constraint.
> > >
> > > I am reminded of the concern as I see that this assumption isn't addressed in the synthetic experiment: for a given game with fixed $n_{att}$, you fix segment lengths to be $k / n_{att}$. What if instead you determined segment lengths based on the information constraints of the game (i.e. make an optimal code using your alphabet and $n_{att}, n_{val}$), and then padded / interleaved the concatenated outputs with random draws from the alphabet?
> > >
> > > (Background assumption: for a communication game with low $n_{att}$, the speaker agent has no pressure to use all of the output characters to communicate its referent. So we shouldn't have any clear expectations over what meaningful content there will be when the speaker is forced to output messages of a certain length.)
> > >
> > > Please let me know if this makes sense / if you agree with the concern and background assumption / if you think the experiment adjustment is worth it!

---

> > > > ### Author Response · Authors · 2022-08-04
> > > > **Additional Response to Reviewer KNRm**
> > > >
> > > > Thank you for reading our response! First, let us address your second comment on the follow-up experiment and then your first comment on the fixed-length.
> > > >
> > > > > I am reminded of the concern as I see that this assumption isn't addressed in the synthetic experiment: for a given game with fixed $n_{att}$, you fix segment lengths to be $k/n_{att}$. What if instead you determined segment lengths based on the information constraints of the game (i.e. make an optimal code using your alphabet and $n_{att},n_{val}$), and then padded / interleaved the concatenated outputs with random draws from the alphabet?
> > > >
> > > > This is an interesting idea!
> > > > **Our concern is, however, that the optimal code would be too “optimal” to be natural.**
> > > > Recall that HAS is based on the jittering behavior of branching entropy $h$.
> > > > That is, HAS assumes that character sequences repeatedly become redundant (the decrease of $h$) and less redundant (the increase of $h$).
> > > > So HAS does not cover the optimal code, as it has no redundancy (i.e., $h$ would be constant).
> > > > **We believe that this is not the fault of HAS**, considering natural languages are, by nature, redundant to some extent.
> > > >
> > > > > I was concerned (but did not write earlier) that the failure of C1 may be due to the fact that the speaker agent is forced to output a fixed-length message, regardless of the amount of information necessary to convey for a particular game. Thus the HAS algorithm picks up on hypo-boundaries of speaker outputs that may be meaningless to the listener but are there due to the output length constraint.
> > > >
> > > > This is another essential question.
> > > > The speaker might output “near-optimal” (instead of “optimal”) code that HAS should recognize and fill the rest with meaningless symbols.
> > > > **We must admit that HAS might fail in such situations** (it becomes more obvious when $k=100,1000,\ldots$).
> > > > **That has to be clarified in the discussion section in the revised version of the paper.**
> > > >
> > > > However, **we emphasize that we made a finding about the property of emergent languages in a standard setting** (i.e., fixed-length messages) from a novel perspective.
> > > > To study in which setting emergent languages satisfy HAS is left for future work.
> > > > Besides, a few studies pointed out that the speaker tends to output meaningless symbols even when the length is variable without any ad-hoc auxiliary loss [Chaabouni+, 2019; Rita+, 2020].
> > > > **Suppressing such an artifact is also crucial in more general emergent communication literature.**
> > > >
> > > > # Reference
> > > > [Chaabouni+, 2019] Rahma Chaabouni, Eugene Kharitonov, Emmanuel Dupoux, & Marco Baroni. “Anti-efficient encoding in emergent communication.” NeurIPS 2019.\
> > > > [Rita+, 2020] Mathieu Rita, Rahma Chaabouni, & Emmanuel Dupoux. “"LazImpa": Lazy and Impatient neural agents learn to communicate efficiently.” CoNLL 2020.

---

### Meta-Review · Area_Chair_4BpQ · 2022-08-29

**Recommendation:** Reject
**Confidence:** Less certain

**Metareview:**

Harris' hypothesis (or articulation scheme, "HAS"; Harris, 1955) suggests that the linguistic boundaries of words can be detected by a count that is unrelated to meaning. In this paper, the authors explore if emergent languages between agents share this property with natural languages. They test the predictions of HAS on the messages produced by converged agents trained in a Lewis signaling game, where a speaker agent is trained by reinforcement learning and a listener agent is trained by supervised learning.
The results are mixed (they find only weak partial support for these predictions in the resulting emergent languages) but the idea is novel and interesting.
All reviewers suggest acceptance, but the paper is IMO just below the decision boundary. It is likely to be of interest to only a small number of people and it is unclear whether it will spark any follow-on work since the Harris' hypothesis doesn't seem to fully hold.

**Award:**

No

---

### Decision · Program_Chairs · 2022-09-14

Reject